# Sleep phenotyping in a rat model of susceptibility to substance use disorders

Eleonora Ficiarà[1,2☉], Oluwatomisin O. Faniyan[2,3☉], Reyila Simayi[2,3], Federico Del Gallo[1], Marisa Roberto[4], Roberto Ciccocioppo[1,2], Michele Bellesi[2,5], Luisa de Vivo[1,2*☉]

1 School of Pharmacy, University of Camerino, Italy, 2 Center for Neuroscience, University of Camerino, 3 International School of Advanced Studies, University of Camerino, Italy, 4 Department of Molecular Medicine, The Scripps Research Institute, La Jolla, California, United States of America, 5 School of Bioscience and Veterinary Medicine, University of Camerino, Italy

☉ These authors contributed equally to the work.
* luisa.devivo@unicam.it

## Abstract

Alcohol use disorders (AUD) are bidirectionally associated with significant sleep disturbances, yet the underlying neural mechanisms remain poorly understood. The Marchigian Sardinian alcohol Preferring (msP) rat is a validated preclinical model that mirrors several genetic and behavioral traits of patients with AUD. This study aimed to characterize the sleep-wake architecture and EEG spectral activity in naïve msP rats compared to Wistar controls. We performed 24-hour polysomnography recordings, revealing that male msP rats (n = 9) spent 7.5% more time awake and less time in NREM sleep relative to Wistar rats (n = 9). This was accompanied by a more fragmented sleep-wake pattern, with a higher number of waking and sleep episodes, state transitions, and sleep fragmentation index. Spectral analysis demonstrated lower high-frequency power, with significant reductions in sigma and beta power during NREM sleep and increased theta/beta ratios during wakefulness. Slow-wave activity, an indicator of sleep pressure, showed a blunted rise and fall across the sleep cycle in msP rats, with reduced amplitude and slope of slow waves during early sleep. Moreover, msP rats exhibited decreased spindle activity, with significantly lower spindle incidence, amplitude, and duration. These findings suggest that msP rats display significant sleep disturbances, including disrupted NREM sleep and altered spectral characteristics in brain activity that partially resemble changes reported in individuals with AUD. This altered sleep profile may reflect neural circuit dysfunctions linked to substance use vulnerability, offering potential insights into the neurobiological basis of sleep disturbances in these complex neuropsychiatric disorders.

## Introduction

Sleep disturbances and alcohol use disorders (AUD) are closely and bidirectionally associated [1–3]. While alcohol negatively impacts sleep architecture and quality

**Data availability statement:** The anonymized raw dataset of EEG and EMG recordings used in this publication has been deposited in Zenodo (DOI: 10.5281/zenodo.15295870) at the following link https://zenodo.org/records/15295870.

**Funding:** This study was supported by the Giovanni Armenise Harvard Foundation Career Development Award to LdV, and the National Institutes of Health grant AA017447 to MR and RC. There was no additional external funding received for this study.

**Competing interests:** The authors have declared that no competing interests exist.

[4–7], many people use it as a form of self-medication to cope with insomnia and other sleep issues. Conversely, sleep problems and insomnia are prevalent among individuals with AUD [8,9] and further exacerbate alcohol consumption and relapse during abstinence [10,11]. Generally, chronic alcohol use is associated with reduced slow wave sleep, increased latency to fall asleep, and reduced sleep continuity, with increased duration of wakefulness after sleep onset (WASO), and a state of general hyperarousal during sleep [12,13]. Moreover, prospective studies suggest that alterations in multiple sleep dimensions (e.g., short duration, eveningness, daytime sleepiness) increase the risk of developing AUD [14–16].

Therefore, understanding the relationship between alcohol drinking and sleep disturbances is crucial for elucidating the mechanisms underlying AUD and its comorbidities, and it can help clarifying whether innate sleep disturbances may predispose to excessive alcohol intake.

Preclinical models of AUD provide a valuable tool for achieving this goal and to develop novel strategies to improve health. The marchigian sardinian alcohol-preferring (msP) rat line is an established model of genetic predisposition to alcohol abuse and has been extensively used to model and investigate the neurobiology of AUD [17,18]. This rat model has proven to be relevant in a translational perspective recapitulating important features seen in individuals with AUD at the behavioral, genetic and brain level. At the behavioral level, msP rats show an anxious and depressive phenotype, which is reduced upon alcohol drinking [19,20]. MsP rats are highly motivated to lever press for alcohol and show high level of seeking behavior in response to stress and environmental cues, supporting the hypothesis that this alcohol-preferring rat line resembles a specific subgroup of AUD patients who drink to alleviate their anxiety and stress [21,22]. Interestingly, in a previous behavioral study it was observed that msP rats have shorter sleeping bouts duration compared to Wistars suggesting a hyperarousal phenotype [23].

At the genetic level, both msP rats and humans share common genetic factors that predispose them to alcoholism. Some of the most compelling findings in msP rats are the strong correlation between two single nucleotide polymorphisms (SNPs) located in the promoter region of the gene encoding the corticotropin-releasing factor 1 (CRF1) receptor, and that msP rats exhibit elevated expression of CRF1 receptor mRNA and protein levels in various brain regions [24]. Interestingly, research has also identified similar polymorphisms in the promoter region of the CRF1 receptor gene in humans, which are associated with AUD [25,26].

In the brain, several neuroimaging studies suggested that some of the morphological and functional features that are observed in msP rats resemble characteristics of AUD patients, such as an innate reduction of grey matter volume in different thalamic and cortical regions, ventral tegmental area, and substantia nigra, and displayed a reduced resting brain function in limbic areas [27,28].

In the light of these findings, we reasoned that a comprehensive characterization of sleep/wake patterns and global brain activity in alcohol-naïve msP rats could add information to further expand the face and construct validity of this preclinical model, informing on the role played by inborn sleep disturbances on the propensity to develop AUD.

To fill this knowledge gap, we performed polysomnographic recordings and an in-depth phenotyping of sleep microarchitecture, characterization of electroencephalogram (EEG) power spectra, slow waves, and sleep spindles in alcohol-naïve msP and Wistar rats. We hypothesized that alcohol preferring msP rats would display an altered sleep/wake cycle relative to control rats, with features similar to those reported in individuals with AUD.

## Materials and methods

### Animals

MsP rats were originally derived from Wistar rats. They have been selectively bred for high alcohol preference and consumption at the University of Camerino (Marche, Italy) for more than 80 generations, beginning from the 13th generation of Sardinian alcohol-preferring (sP) rats that were originally developed at the University of Cagliari (Sardinia, Italy) [29]. Wistar (n = 9 male) and msP (n = 9 male) rats were bred at the University of Camerino, Italy. Rats were housed in groups of 3–4 animals per cage, on a 12h light/dark cycle (lights on at 8 AM), in a temperature (20–22°C) and humidity (45–50%) controlled room. Food and water were provided ad libitum. Procedures were conducted in adherence with the European Community Council Directive for Care and Use of Laboratory Animals and the National Institutes of Health Guide for the Care and Use of Laboratory Animals (authorization 542/2023-PR, prot. 1D580.40).

### Polysomnographic recordings

Rats were anesthetized with gaseous isoflurane (3% for induction, 1–1.5% for maintenance) and placed in a stereotaxic apparatus. Lubricant eye ointment was locally applied, and rats were administered 5 mg/kg carprofen subcutaneously. The body temperature was kept at 37–38°C by a thermostatic blanket; heart rate, breath rate and oxygen saturation were monitored during the whole procedure. Rats' responses to tail and limb pinch were checked periodically throughout the procedure to ensure the animals remained unconscious. After skin disinfection, the skull was exposed and rats were implanted for chronic EEG recordings with electrodes over the frontal cortex (from bregma: anteroposterior +2 mm, mediolateral bilaterally, ± 2 mm), the parietal cortex (anteroposterior, −2 mm; mediolateral, ± 3 mm), and cerebellum (both reference and ground electrode). For electromyographic (EMG) recordings, two stainless steel wires were placed into neck muscles. Finally, if necessary, the skin around the implant was sutured and further disinfected. Hydration during the procedure was ensured by injecting saline subcutaneously. After surgery, rats were monitored until they regained consciousness and then they were housed individually for 7 days to allow full recovery. Subsequently, rats were moved to the recording chamber, connected to a Pre-Amplifier (Intan Technologies, RHD 16-Channels) and to the Open Ephys acquisition board [30] and allowed to habituate for 48 hours before recording 24 hours of baseline. EEG and EMG signals were sampled at 1Khz using the open access Open Ephys Acquisition Software (GUI) and filtered (EEG: high-pass filter at 0.1 Hz; low-pass filter at 40 Hz; EMG: high-pass filter at 10 Hz; low-pass filter at 500 Hz). All data were down sampled at 512 Hz for off-line manual scoring (Sleep Sign, Kissei Comtec Co., Ltd) of individual rats which occurred in 4s epochs following standard criteria [31]. Epochs with artifacts, primarily occurring during active wakefulness, were excluded from the spectral analysis. EEG/EMG recordings at baseline occurred throughout a similar age interval for both groups (post-natal day, P, 58 ± 8 days Wistar, 58 ± 12 days msP, Shapiro-Wilk test Wistar p = 0.678, msP p = 0.47, unpaired t-test p > 0.99, S1 Fig). At the end of the recordings, rats were euthanized with carbon dioxide.

### Sleep architecture and quantity

The time spent in each vigilance state (Wake, NREM sleep, REM sleep) and the number and duration of wake, NREM and REM sleep episodes, defined as lasting for 8 seconds (s) or more, were quantified. Sleep attempts were defined as sleep episodes <20 s preceded and followed by wake episodes >20s. Brief awakenings were defined as short episodes of arousal <16 s [32].

NREM sleep onset was defined as the time elapsed between lights ON and the first consolidated NREM sleep episode, lasting at least 20 seconds. The number of transitions between vigilance stages (Wake-NREM, Wake-REM, NREM-Wake, NREM-REM, REM-Wake, REM-NREM) was also calculated.

### Spectral analysis

EEG power spectra were computed by a fast Fourier transform (FFT) routine for 4-s epochs with 0.25 Hz resolution both for parietal and frontal channels. Epochs containing EEG artifacts recognized during visual scoring were excluded from spectral analysis.

Power spectra for the following frequency bands were also computed: delta and Slow Wave Activity (SWA): 0.5–4 Hz; Low Theta: 5–8 Hz; High Theta: 8–12 Hz; Sigma: 12–15 Hz in NREM sleep; Beta: 15–30 Hz; Theta/Beta: 5–12/ 15–30 Hz. One msP rat was excluded from the analysis of beta power due to wrong settings of the low pass filter during signal recording.

### Motor activity

Wake epochs were further classified offline as quiet or active wake periods according to the amount of EMG activity, as in [33]: all wake epochs showing a power below and above a threshold were classified as quiet wake and active wake, respectively. The threshold was determined by calculating the 95th percentile of EMG activity during all NREM sleep episodes; this threshold ensured that the identified "quiet wake" epochs were only those that contained minimal amounts of EMG activity.

### Slow waves and spindles detection

During each NREM sleep period within the first 3 hours of the light period (early sleep) and the last three hours of the light period (late sleep), single slow waves and spindles were identified using the YASA algorithm [34], with the following parameters set:

- Slow waves frequency 0.5-4 Hz; duration of the negative deflection of the slow wave 0.1-2 s; duration of the positive deflection of the slow wave 0.1-2 s; negative peak amplitude 10-400 µV; positive peak amplitude 10-400 µV; minimum peak-to-peak (PTP) amplitude 30 µV.

- Spindles frequency range 12-18 Hz; detection threshold: moving correlation between original signal and sigma-filtered signal: 0.75, number of standard deviations above the mean of a moving root mean square of sigma-filtered signal 1.5; duration of the spindle 0.5- 15 s.

### Statistical analysis

Data were analyzed using GraphPad (version 10.2.2), custom-made Matlab (version R2023a) and Python (version 3.10) codes.

Normality of data was evaluated using the Shapiro–Wilk test, alongside inspection of QQ plots and residuals. For datasets that violated normality assumptions, nonparametric tests were used; otherwise, parametric tests were conducted. The presence of outliers was tested using Grubbs' test (alpha = 0.05), most sensitive to single outliers, and ROUT test (False Discovery Rate Q = 1%), which applies non-linear regression to flag multiple outliers while controlling for false discovery rate [35]. When outliers were identified, they were excluded from subsequent analyses. In case of missing values, data were analyzed by fitting mixed-effects models. S1 Table provides a summary of outliers found and the statistical tests applied.

## Results

### Sleep/Wake time and architecture

Across the 24 hours, msP rats spent 7.5% more time awake and less time in NREM sleep relative to Wistar rats (Fig 1A-1C; msP: 726.6 ± 44.3 minutes in wake, 566.2 ± 42.2 in NREM sleep, 145.9 ± 37.0 in REM sleep; Wistars: 683.8 ± 42.6

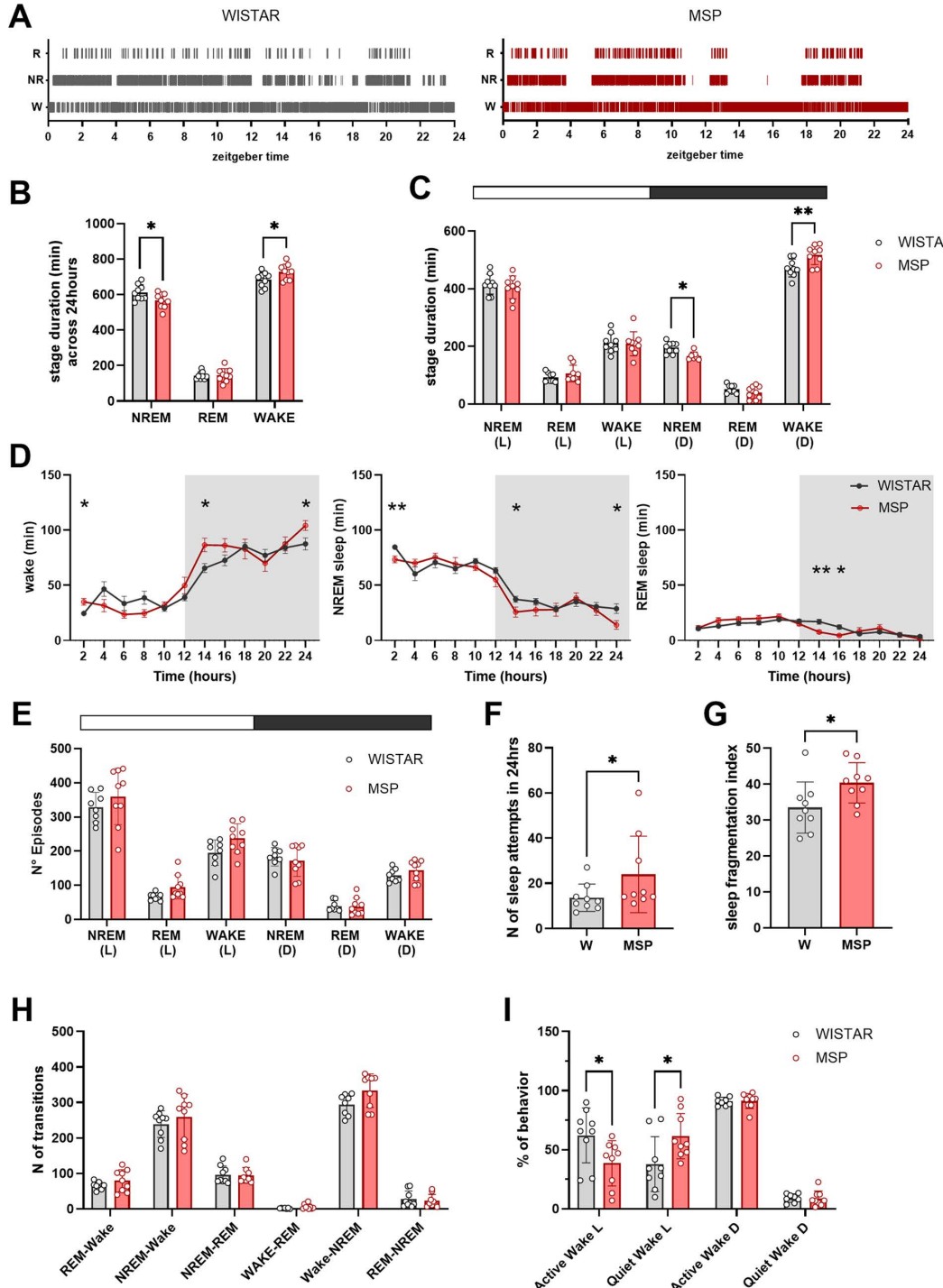

**Fig 1. Sleep architecture in Wistar and msP. (A)** Representative hypnograms for Wistar and msP rats. **(B-C)** Distribution of wake, NREM sleep, and REM sleep amount (mean±std) in the 24h (B) and during light and dark periods **(C)**. **(D)** Twenty-four-hour time course of minutes spent in each state (mean±SEM). Grey areas indicate the dark cycle, from 8:00 pm to 8:00am. **(E)** Distribution of the number of episodes in the light, L, and dark, D, phase (mean±std). **(F)** Number of sleep attempts during the 24 hours. **(G)** Sleep fragmentation index computed as the number of wakefulness episodes/ total sleep time. **(H)** Number of transitions between stages. **(I)** Percentage of time spent in active or quiet wake during the light, L, and dark, D, period. Empty circles represent individual rats (Wistar n = 9, msP n = 9). * p < 0.05, ** p < 0.01.

minutes in wake, 612.3±43.6 in NREM sleep, 143.8±20.6 minutes in REM sleep; Fig 1B, two-way ANOVA, interaction *stage x rat line* F(2, 48) = 5.793, p=0.0056, Fisher's LSD post-hoc test msP vs Wistar wake p=0.0247, NREM sleep p=0.0163; Fig 1C, two-way ANOVA, *stage x rat line* F(5, 95)=3.9 p=0.0057, NREM dark phase p=0.046, Wake dark phase p=0.0013). A more detailed analysis showed that msP rats spent more time awake in the 2 first and last hours of the dark phase as well as in the first 2 hours of the light phase (Fig 1D, mixed effects models, wake: interaction *time x rat line* F(11, 190) = 2.63, p=0.038, *rat line* F(1, 190) = 1.34 p=0.25. NREM sleep: *rat line* F(1, 191) = 3.89, p=0.05, interaction *time x rat line* F(11, 191) = 1.63, p=0.09. REM sleep: interaction *time x rat line* F(11, 175) = 3 p=0.0011, *rat line* p=0.93, uncorrected Fisher's LSD post hoc test), although this effect disappears after correcting for multiple comparisons. Moreover, we found that msP rats had a more fragmented wake/sleep pattern, as highlighted by the overall higher number of episodes (Fig 1E, two-way ANOVA, *rat line* F(1,90) = 4.68 p=0.033, interaction *time x rat line* F(5,90)=1.13, p=0.35), the higher number of sleep attempts across the 24 hours (Fig 1F, Mann-Whitney p=0.0468), the higher sleep fragmentation index (Fig 1G unpaired t-test p=0.036), and the trend for a higher number of state transitions between strains (Fig 1H, mixed effect model, *rat line* F(1, 92) = 3.67 p=0.058, interaction *transition x rat line* F(5, 92) = 1.16 p=0.33). Average duration of waking and sleep bouts was not significantly different between msP and Wistars and we found no difference in NREM sleep latency between msP and Wistars (not shown).

To further characterize msP activity, we assessed the amount of quiet and active wakefulness, by using an index of motor activity based on the 4 sec behavioral staging and the power of the EMG signal [33]. We found that msP rats spent more time in quiet waking during the light phase relative to Wistar rats whereas no differences were found during the dark phase (Fig 1I, % of quiet waking relative to total waking: Wistar=37.90%, msP=61.44%. mixed effect model, interaction *state x rat line* F(3, 62)=6.76 p=0.0005, *state* F(1.094, 22.61)=77.56, p<0.0001, *rat line* F(1,62)=3x10^{-17} p=0.99, Sidak multiple comparisons test: Wistar vs msP in quiet waking during the light period p=0.032, in active wake during the light period p=0.032, see S1 Table for other comparisons).

In summary, undisturbed msP rats spent more time awake and had a more fragmented sleep-wake pattern relative to Wistar rats.

## Spectral composition of vigilance states

In the frontal and parietal derivations (Figs 2-3), absolute EEG spectral power of the high frequency range was lower in msP rats across all vigilance states. Specifically, while average slow wave activity (SWA) was similar between the two rat lines (p=0.26 unpaired t-test), absolute sigma and beta power in NREM sleep were reduced by more than 50% in msP rats relative to Wistars (p<0.0001) (Fig 2A). During REM sleep, low-theta power tended to be higher in msP rats (p=0.078), high theta did not differ, and beta power was significantly lower relative to Wistar (p=0.0009) (Fig 2B). During wakefulness, in msP rats, high theta and beta power were reduced (high theta p=0.0116, beta p=0.0011), whereas the ratio total theta/beta power was higher relative to Wistar (p=0.0082) (Fig 2C). We observed equivalent results in the parietal derivations (Fig 3). Beta power was also reduced during brief awakenings (p<0.0001).

## Sleep slow waves

SWA progressively increases with the duration of wakefulness, reaches its peak at the onset of sleep, and gradually decreases during sleep. Since SWA is considered a measure of sleep intensity and a marker of the homeostatic sleep process [36], we aimed to compare SWA changes throughout 24-hours of undisturbed recordings in Wistar and msP rats. SWA computed in 1-hour intervals revealed that msP rats exhibited a diminished rise and decline relative to Wistar (Fig 4A, mixed model, interaction *rat line x time* F (23,354) = 2.73 p<0.0001). The difference in SWA between the initial and final hours of the light phase, as well as between the first and last hours of the dark period, was significantly lower in msP rats compared to Wistar rats (Fig 4B-4C, unpaired t-test light period p=0.0042, dark period p=0.015).

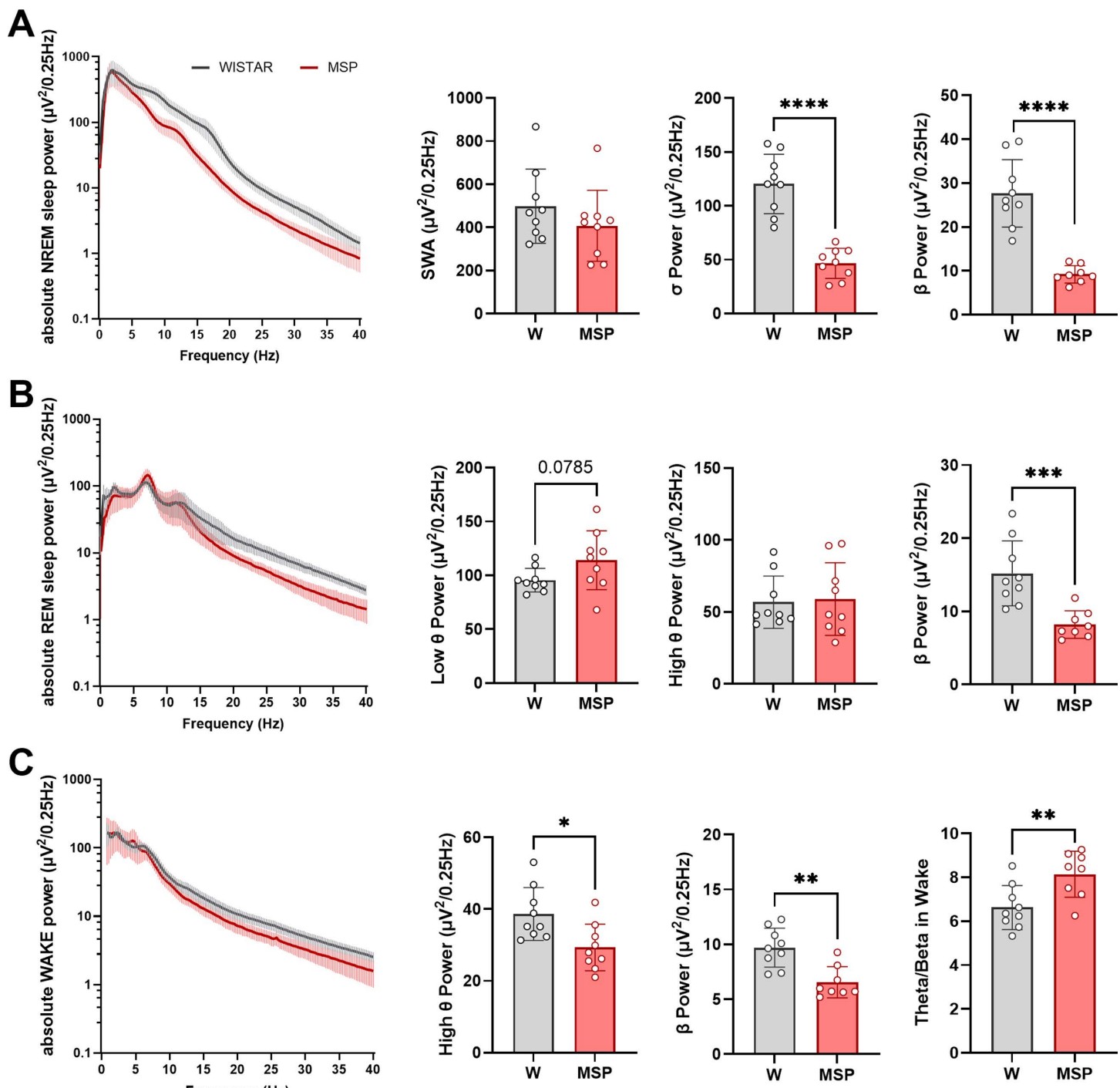

**Fig 2. EEG power spectrum of frontal channels. (A)** Power spectrum of EEG (mean±std) during NREM sleep and values of SWA (0.5-4 Hz), NREM sleep sigma (12-15 Hz) and beta (15-30 Hz) power across 24 hours. **(B)** Power spectrum of EEG during REM sleep and values of REM sleep low theta (5-8 Hz), high theta (8-12 Hz) and beta power across 24 hours. **(C)** Power spectrum of EEG during wake and values of high theta, beta, and theta/beta ratio power across 24 hours during waking epochs. Values are mean±std. Empty circles represent values for individual rats. * p<0.05; ** p<0.01; *** p<0.001; **** p<0.0001.

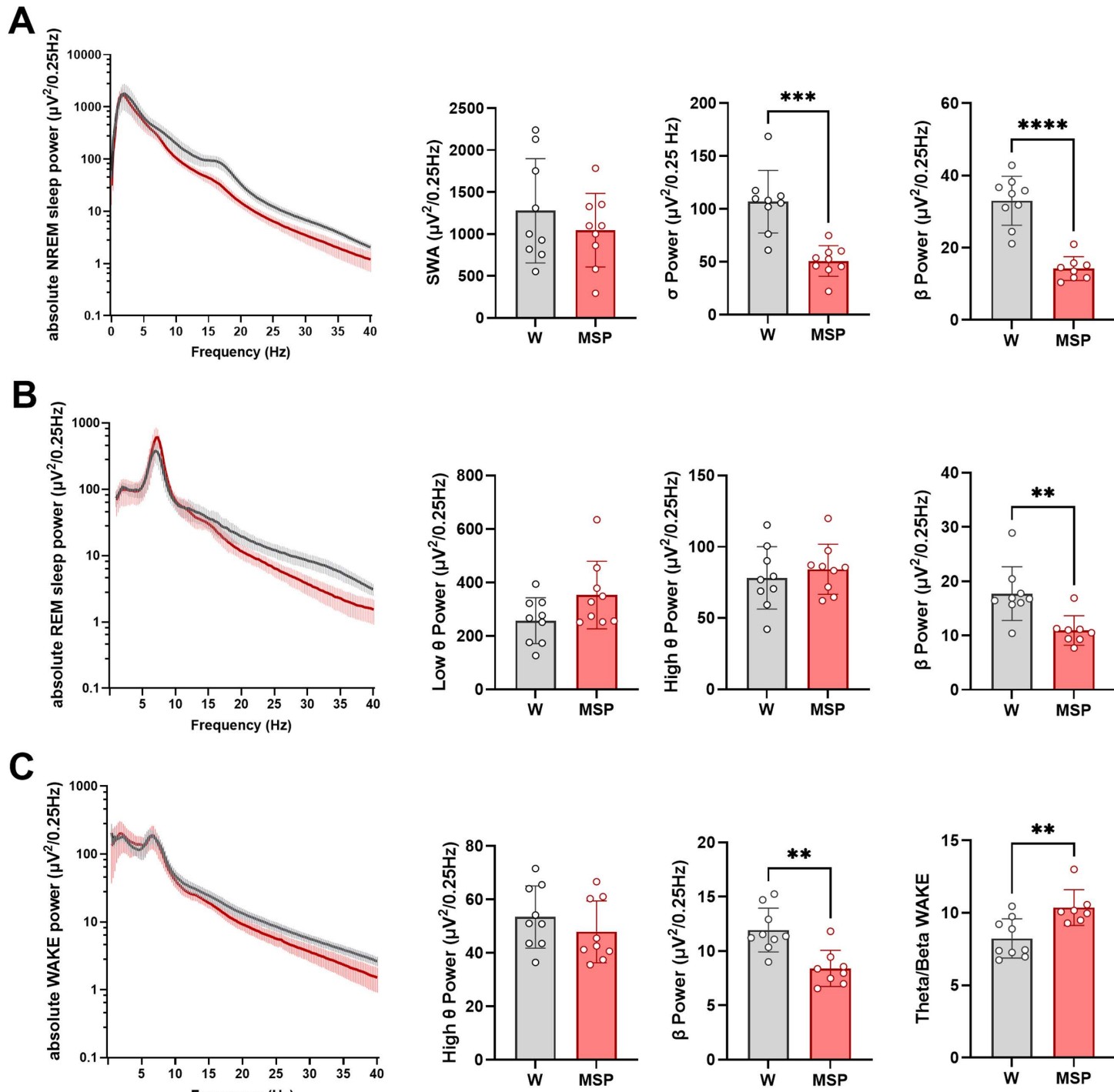

**Fig 3. EEG power spectrum of parietal channels. (A)** Power spectrum of EEG (mean ± std) during NREM sleep and values of SWA (0.5-4 Hz, p = 0.37 unpaired t-test), NREM sleep sigma (12-15 Hz, p = 0.0001) and beta (15-30 Hz, p < 0.0001) power across 24 hours. **(B)** Power spectrum of EEG during REM sleep and values of REM sleep low theta (5-8 Hz, p = 0.14, Mann-Whitney test), high theta (8-12 Hz p = 0.52, unpaired t-test) and beta (p = 0.0036, unpaired t-test) power across 24 hours. **(C)** Power spectrum of EEG during wake and values of high theta (8-12 Hz, p = 0.32, Mann-Whitney test), beta (p = 0.0014, unpaired t-test) and theta/beta ratio power across 24 hours during waking epochs (p = 0.0037, Mann-Whitney test). Values are mean ± std. Empty circles represent individual rats. * p < 0.05; ** p < 0.01; *** p < 0.001; **** p < 0.0001.

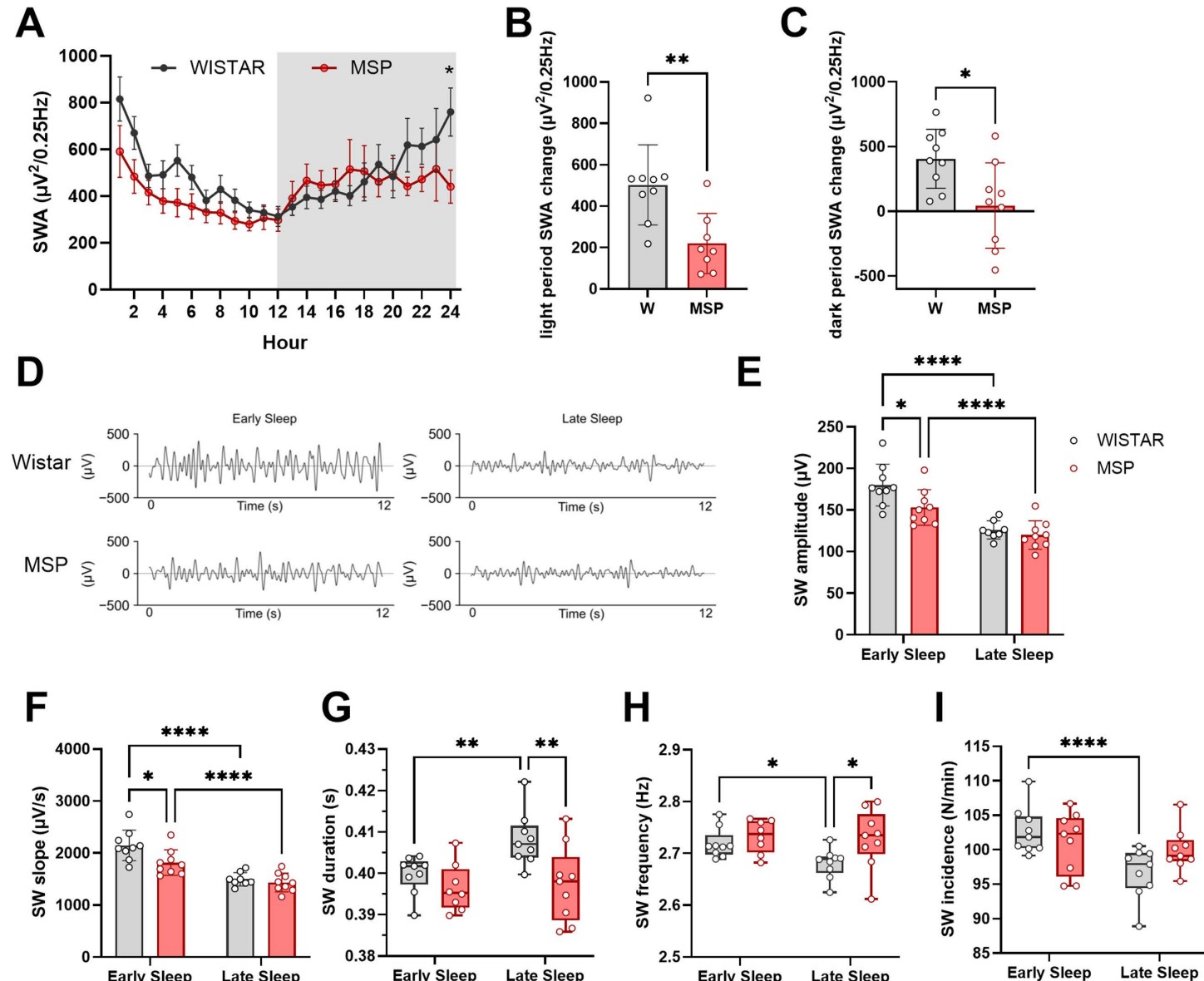

**Fig 4. SWA and sleep slow waves. (A)** Twenty-four-hour time course of SWA (mean±sem), two-way ANOVA interaction rat line x time $p < 0.0001$, uncorrected Fisher LSD post hoc test $p = 0.02$. The grey area indicates 12hrs of lights off. **(B)** SWA change during the light period (1st hour – 12th hour of light period, unpaired t-test $p = 0.0042$), mean±std. **(C)** SWA during the dark period (1st hour – 12th hour of dark period, unpaired t-test $p = 0.0154$), mean±std. **(D)** Representative EEG signal (band-pass filtered 0.5-4 Hz) during early and late sleep for Wistar (upper part) and msP (lower part). **(E)** Values of slow wave amplitude during early and late sleep. **(F)** Slope of slow waves during early and late sleep. **(G)** Duration of slow waves. **(H)** Frequency of slow waves during early and late sleep. **(I)** Incidence of slow waves. Columns indicate mean±SEM. Empty circles are single rats. Data in E-I were analyzed with repeated measures two-way ANOVA followed by Sidak's multiple comparison test, see Supplementary Table 1 for detailed statistical results. * $p < 0.05$, ** $p < 0.01$, **** $p < 0.0001$.

Since SWA merges different aspects of slow waves into a single metric, we applied a slow wave detection algorithm to quantify multiple parameters of individual waves. To capture the net change of these features over the course of sleep, we focused the analysis on early (first 3 hours of light period) and late (last 3 hours of light period) NREM sleep (Fig 4D), corresponding to periods of high and low sleep pressure, respectively [37]. In early sleep, msP slow wave amplitude and

slope were smaller relative to Wistar rats (Fig 4E-4F; mixed effect model for *amplitude*, effect of rat line F (1, 16) = 4.3 p = 0.055, time F (1, 15) = 127.6 p < 0.0001, interaction F (1, 15) = 5.2 p = 0.043, Sidak's multiple comparison test msP vs Wistar early sleep p = 0.019, late sleep p = 0.49; for *slope*, effect of rat line F (1, 16) = 4.73 p = 0.045, time F (1, 15) = 126.7 p < 0.0001, interaction F (1, 15) = 6.27 p = 0.024, Sidak's multiple comparison test msP vs Wistar early sleep p = 0.011, late sleep p = 0.53). In late sleep, slow waves had shorter duration in msP relative to Wistars (Fig 4G, S1 Table, mixed effect model interaction *time* x *rat line* F (1, 15) = 6.369 p = 0.0234, *rat line* F (1, 16) = 5.386 p = 0.034, *time* F (1, 15) = 6.129 p = 0.026, Sidak's multiple comparison test late sleep, msP vs Wistar p = 0.0046, early vs late sleep Wistar p = 0.0051, msP p = 0.55). Moreover, slow waves had faster frequency (Fig 4H, S1 Table, mixed effect model interaction *rat line* x *time* F(1,15) = 4.99 p = 0.041, Sidak's multiple comparison test early vs late sleep, Wistar p = 0.0185, msP p = 0.96, in late sleep Wistar vs msP p = 0.034) and were more numerous in early than in late sleep in Wistar rats but not in msP (Fig 4I, S1 Table, repeated measure two-way ANOVA, interaction *time* x *rat line* F (1,16) = 13.55 p = 0.002, Sidak's multiple comparison test early vs late sleep Wistar p < 0.0001, msP p = 0.45).

### Sleep spindles

Mean power values in the sigma range exhibited a trend opposite to that of SWA during sleep, gradually increasing from early to late sleep during the light phase (Fig 5A). However, we found that msP showed a significant lower amount of NREM sigma power throughout the 24 hours and a blunted change from early to late sleep (Fig 5A-5B). Sigma power is widely used to describe at the EEG level global spindle quantity and properties both in humans and rodents [38]. Sleep spindles are trains of distinct waves occurring during NREM sleep bouts, with a frequency that, in rodents, usually spans the 10–20 Hz range with a duration ≥0.5 seconds. They reflect the activity of cortico-thalamocortical circuits and are thought to play a crucial role in protecting NREM sleep from sensory or spontaneous disruption, and in facilitating memory formation. Since we found lower sigma power and blunted dynamics during sleep in msP relative to Wistar rats, we used an algorithm to identify single spindle events to study their characteristics over the course of sleep (Fig 5C-5H). Spindle detection and quantification revealed that all spindle features (incidence, frequency, amplitude, duration, and number of oscillations) were reduced in msP rats both in early and in late sleep relative to Wistar rats (mixed effect model for *amplitude*, effect of rat line F (1,16) = 45.15 p < 0.0001, time F (1,15) = 3.66 p = 0.07, interaction time x rat line F (1, 15) = 1.385 p = 0.46; two-way repeated measure ANOVA for *incidence*, effect of rat line F (1, 16) = 23.68 p = 0.0002, time F (1, 16) = 124.8 p < 0.0001, interaction time x rat line F (1, 16) = 26.09 p = 0.0001, Sidak's multiple comparison test msP vs Wistar early sleep p = 0.0142, late sleep p < 0.0001; for *frequency*, effect of rat line F (1, 16) = 15.69 p = 0.0011, time F (1, 16) = 0.29 p = 0.60, interaction time x rat line F (1, 16) = 2.74 p = 0.12, Sidak's multiple comparison test msP vs Wistar early sleep p = 0.0084, late sleep p = 0.0003; mixed effect model for *duration*, effect of rat line F (1, 16) = 22.52 p = 0.0002, time F (1,15) = 15.08 p = 0.0015, interaction F (1,15) = 2.91 p = 0.1; for *number of oscillations*, effect of rat line F (1,16) = 25.8 p = 0.0001, time F (1,16) = 8.3 p = 0.011, interaction F (1,16) = 0.97 p = 0.3; see S1 Table for details on multiple comparisons), as well as during the other 6 hours of the light period (S2 Fig).

### Discussion

In this work we analyzed 24 hours of baseline EEG recordings, investigating motor activity, sleep architecture, spectral composition and main characteristics of slow waves and spindles events during NREM sleep in msP and Wistar rats.

We found that undisturbed alcohol-naïve msP rats spent more time awake, especially during the dark period, and prefer quiet rather than active waking during the light period, relative to Wistars. Additionally, msP rats displayed a more fragmented wake/sleep pattern, as highlighted by the higher number of waking, NREM and REM sleep episodes, especially during the light period, the higher number of sleep attempts and state transitions across 24 hours, and the higher sleep fragmentation index relative to Wistar rats. However, the latency to NREM sleep onset was similar between rat lines.

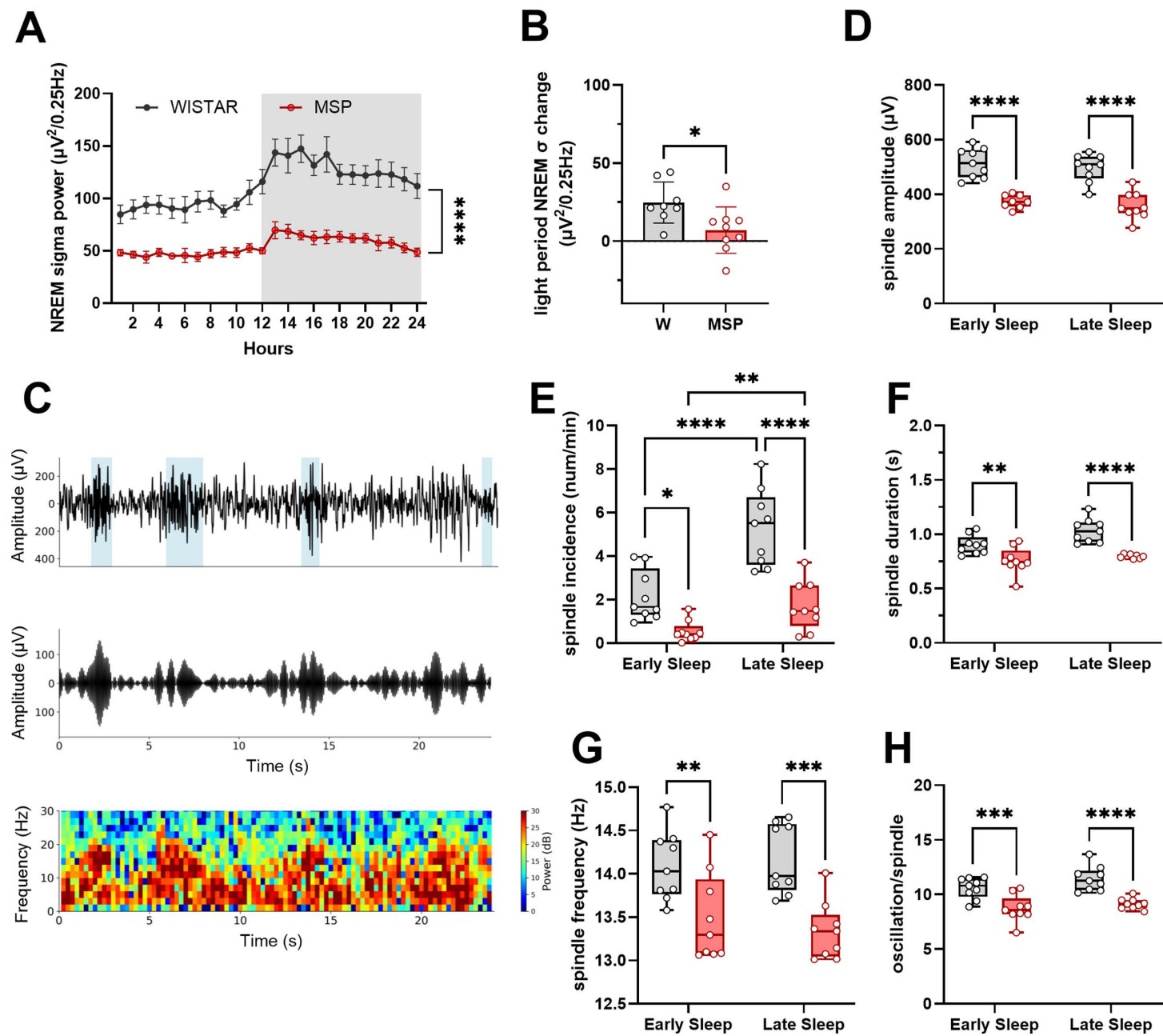

**Fig 5. Sigma and spindle activity. (A)** Twenty-four-hour time course of NREM sigma power (mean±SEM). Two-way ANOVA, effect of rat line p<0.0001, interaction time x rat line p<0.0001. **(B)** NREM sigma change during the light period (mean at 12hrs – mean in the 1st hour), unpaired t-test p=0.021. **(C)** Representative trace of EEG signal from Wistar rat (raw top, filtered12-15 Hz middle) and spectrogram (bottom). **(D)** Values of spindle amplitude during early and late sleep. **(E)** Incidence of sleep spindles (spindles/min) in early and late sleep. **(F)** Length of -spindles increases in Wistar rats from early to late sleep, but not in msP. **(G)** Frequency of spindles peaks around 14 Hz in Wistars, at 13.3 Hz in msP. **(H)** Number of oscillations in each spindle is higher in Wistar rats relative to msP. Wistar (grey); msP (red). Empty circles represent individual rats. Sidak's multiple comparisons test. * p<0.05; ** p<0.01; *** p<0.001; **** p<0.0001.

Increase waking and sleep fragmentation are consistent with an overactivity of the corticotropin-releasing factor (CRF) system, reported in different brain regions of msP rats, and ascribed to two single-nucleotide polymorphisms (SNPs) at the CRF1 receptor locus [24,39]. CRF is known to act as a neurotransmitter that contributes to general cortical arousal

and autonomic nervous system activation. Several lines of evidence support the role of CRF in the regulation and/or modulation of spontaneous waking [40,41]. CRF suppresses slow wave sleep through activation of the hypothalamic-pituitary-adrenal (HPA) system and inhibition of the sleep-promoting growth hormone-releasing hormone [42] and stress induced sleep fragmentation in mice is prevented by administration of CRF1 receptor antagonist [43]. However, in msP we did not observe changes in the amount of REM sleep, a suggested biomarker for clinical conditions associated with enhanced CRF secretion [44]. This suggests that CRF changes in this rat line may not play a major role in the maintenance and/or generation of REM sleep. This is consistent with earlier studies showing that administration of the glucocorticoid antagonist mifepristone did not restore the alterations in sleep bouts duration observed in msP rats [23].

The increased amount of waking in msP was not accompanied by a corresponding increase in motor activity. During the light phase, msP rats spent more time in quiet-waking than in active-waking and displayed overall more quiet-waking than Wistar rats. This difference was not present during the dark phase when time spent in active wake was comparable between the two rat lines.

The sleep/wake pattern fragmentation observed in msP rats could also be linked to the differential expression of aldehyde dehydrogenase 2 (Aldh2) encoding gene between msP and Wistar rats. Genetic mapping of the msP line with microarrays and gene sequencing [21,45] showed that several genes that encode aldehyde dehydrogenase isoforms are altered in msP rats compared with unselected Wistar controls. For instance, the Aldh2 gene was downregulated, and Aldh1a1, Aldh1a4, Aldh3a2, and Aldh5a were upregulated. Studies in humans have identified an association between Aldh2 deficiency and the rs671(A) allele, linking it with symptoms such as challenges in maintaining sleep, excessive daytime sleepiness, reduced scores on the Mini-Mental State Examination (MMSE), and an increased prevalence of cognitive impairments [46–48]. Additionally, Aldh2 deficiency is thought to cause a buildup of neurotoxic aldehyde metabolites, including those derived from dopamine, serotonin, and noradrenaline. This accumulation may lead to monoaminergic neuronal loss and disrupt the regulation of wakefulness and sleep [49–51]. Indeed, differentially expressed genes related to catecholamine, glutamate, and GABAergic transmission have been detected in msP rats relative to Wistars [21], and may underly some of the alterations in sleep patterns and EEG power spectra observed in this study.

In msP rats, absolute EEG spectral power of the high frequency range was lower while theta/beta ratio (TBR) was higher relative to Wistar rats across all vigilance states. Spontaneous TBR is hypothesized to reflect frontal cortical regulation of subcortical processes [52] and to be a promising electrophysiological marker for prefrontally mediated executive control functions [53]. Numerous studies have demonstrated that TBR is elevated in attention deficit/hyperactivity disorder (ADHD) and that a negative relationship exists between TBR and trait attentional control, as well as attentional orienting [54], emotion regulation [55], and behavioral inhibition [56] in healthy adults, so that individuals with high TBR have difficulties in inhibiting emotional stimuli or regulating their emotions. Thus, even though TBR relevance in rodents has not been confirmed yet, higher TBR in msP rats is consistent with their increased anxiety, stress hypersensitive, and depressive-like phenotype revealed by multiple behavioral paradigms [20,57,58].

SWA is a reliable marker of sleep pressure and process S in mammals, being high at the beginning of sleep, gradually decreasing during sleep and increasing with time spent awake [59]. Moreover, SWA reflects changes in amplitude, incidence and slope of sleep slow waves, which depend on the strength of synaptic connections and other mechanisms such as GABAergic transmission [60]. While we observed no significance change in 24-hour average SWA between msP and Wistar rats, SWA decline and rise across the 24 hours was narrower in msP rats and sleep slow wave amplitude and slope were significantly smaller relative to Wistars. Therefore, the differences observed in msP rats suggest lower accumulation of sleep pressure, neural circuit dysfunction, and/or impaired mechanisms of synaptic plasticity in this rat line, both in wake and during sleep. Interestingly, differences in the expression of molecules important for synaptic plasticity, learning, and memory, such as glutamate, GABA, BDNF, and calcium/calmodulin-dependent protein kinases (CaMKIg and CaMKIIa), have been reported in msP rats [24].

Sigma power and sleep spindles were also drastically lower in msP rats relative to Wistars. Sleep spindles have been linked to brain plasticity and have been shown to be positively associated with cognitive functions such as intelligence and learning ability in humans. Conversely, they are negatively correlated with social anxiety in adolescents [61] and are impaired by alterations of the thalamocortical circuit function. Hence, msP trait-like reduction in spindle activity could reflect deficits in the thalamocortical system and, potentially, in msP ability to acquire and consolidate memory. Future research should determine which cellular mechanisms drive their reduced spindle activity and whether msP rats have impaired learning and long-term memory consolidation relative to Wistar rats.

Interestingly, impaired synaptic plasticity, reduced slow wave and spindle activity have been reported in a variety of neuropsychiatric conditions, including in male patients with major depressive disorders (MDD) [62,63] and in young-sters at risk of developing alcohol use disorders [64,65]. These and our findings suggest that the neurobiological alterations underlying deficits in sleep slow waves and spindles, and possibly in learning, found in msP rats and in youngsters at risk of AUD might constitute an endophenotype which contributes to their propensity for alcohol use. The relationship between cognitive functions and vulnerability to AUD has been studied focusing mostly on cognitive control (such as response inhibition, response selection, flexible updating), decision making, and reward learning (such as reward valuation, discount probability). Reduced cognitive flexibility and top–down, inhibitory control, linked to dysfunction in frontal-subcortical brain circuits, seems to confer vulnerability to AUD [66–68], and attentional bias to reward [69] and maladaptive reward learning are suggested to predict alcohol use progression [70]. A longitudinal study with a 40-year follow-up, identified reduced cognitive efficiency as one of the two main premorbid predictors of both alcohol dependence and failure of remission later in life [71]. Moreover, existing studies show an association between increased risk of AUD and low scores in broad intelligence tests and in cognitive functions encompassing attention and working memory [72], even though conflicting results are present in the literature [73,74]. To this regard, controlled preclinical experiments could be useful to test which cognitive domains are associated with increased risk of AUD, without the confounding factors typical of human studies.

Other alterations of sleep patterns typically found in individuals with MDD, such as increased NREM sleep onset and longer REM sleep, were not found in alcohol-naïve young msP rats. Similarly, lower theta and beta power in alcohol-naïve msP rats do not parallel human reports of increased theta and beta power in individuals with anxiety, hyperarousal, and AUD [75–77]. It is known that GABA neurotransmission is involved in the generation of brain oscillation in the beta and gamma frequencies: for instance, the increase of GABA-A receptors conductance on inhibitory neurons can increase the power of beta oscillations in the sensorimotor cortex [78]. Moreover, polymorphism in GABA receptor genes have been associated both with alterations of EEG beta power and with increased risks of developing AUD [79]. While changes in the GABA system at the molecular and electrophysiological level have been reported in msP, a comprehensive charac-terization of the excitation-inhibition homeostasis has not been conducted yet in this rat line. In vitro electrophysiological studies focused on the Central Amygdala where an upregulation of the inhibitory and excitatory neurotransmission have been reported [39,57,80]. Whether other brain regions (e.g., somatosensory cortices) display a downregulation of GAB-Aergic, cholinergic or noradrenergic neurotransmission which could explain lower EEG beta and theta power in msP rats is not known. Moreover, underlying differences in the organization, connectivity, and functionality of cortical and subcorti-cal circuits between rats and humans might explain such discrepancies.

This study has several limitations, including the lack of alcohol exposure to measure the direct association between sleep EEG features and the amount or pattern of voluntary alcohol drinking, as well as the effects of ethanol on sleep microarchitecture in msP rats. However, extensive research has characterized the effects of acute alcohol consumption on sleep and EEG activity in both individuals with AUD and healthy controls (reviewed in [5,7]). Additionally, one study found that binge alcohol treatment affects sleep-wake activity differently in alcohol-preferring (P) and non-preferring (NP) rats, particularly during the dark period following ethanol exposure [81]. P rats were more susceptible to alcohol, exhibiting a significant reduction in NREM sleep, increased wakefulness, and no change in REM sleep compared to NP rats.

Similarly, the study would have benefited from in vitro electrophysiological recordings and behavioral tests linking the blunted slow wave homeostasis and reduced spindle activity in msP rats with specific deficits in synaptic plasticity, learning, and memory consolidation.

In conclusion, this study provides novel insights into the sleep microarchitecture of alcohol-naïve msP rats, a model known for its genetic and behavioral similarities to patients with AUD. Our findings reveal significant differences in sleep patterns between msP and Wistar rats, particularly in terms of sleep fragmentation, quiet waking, TBR, SWA and spindle activity. However, discrepancies between the msP model and human EEG profiles in AUD (e.g., decreased rather than increased beta power) suggest the need for further research to fully understand the predictive value of these findings.

Future investigations should address the limitations of current study by including both sexes, different ages, exploring the effects of alcohol exposure, and performing longitudinal assessments of behavioral comorbidities to establish whether these EEG alterations can serve as early markers of AUD risk.

## Supporting information

**S1 Table. Summary of statistical results.** Outliers identified by the Grubbs' test were removed from the statistical analysis.
(DOCX)

**S1 Fig. Rat age and slow wave activity. A)** age at baseline recordings for each individual rat. W = Wistars, 58 ± 8 days, msP 58 ± 12 days, unpaired t-test p > 0.99. **B)** mean NREM SWA levels (24 hours) across age from frontal cortex. Each symbol refers to a single animal.
(TIF)

**S2 Fig. Average spindle features during 6hr interval between zeitgeber time 4–9.** A) spindle incidence B) spindle amplitude. C) spindle duration. D) spindle frequency. E) number of oscillations per spindle. W = Wistar, unpaired t-test. ** p < 0.01, ***p < 0.001.
(TIF)

## Author contributions

**Conceptualization:** Michele Bellesi, Luisa de Vivo.

**Data curation:** Eleonora Ficiarà, Oluwatomisin O. Faniyan.

**Formal analysis:** Eleonora Ficiarà.

**Funding acquisition:** Marisa Roberto, Roberto Ciccocioppo, Luisa de Vivo.

**Investigation:** Oluwatomisin O. Faniyan, Reyila Simayi, Federico Del Gallo.

**Project administration:** Luisa de Vivo.

**Resources:** Roberto Ciccocioppo, Luisa de Vivo.

**Software:** Eleonora Ficiarà, Federico Del Gallo.

**Supervision:** Michele Bellesi, Luisa de Vivo.

**Validation:** Eleonora Ficiarà, Oluwatomisin O. Faniyan, Federico Del Gallo.

**Visualization:** Luisa de Vivo.

**Writing – original draft:** Eleonora Ficiarà, Luisa de Vivo.

**Writing – review & editing:** Marisa Roberto, Roberto Ciccocioppo, Michele Bellesi, Luisa de Vivo.

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
