## [Decision Letter · Decision Letter 0]

29 Dec 2024

PONE-D-24-48364Sleep phenotyping in a rat model of susceptibility to substance use disordersPLOS ONE

Dear Dr. de Vivo,

Thank you for submitting your manuscript to PLOS ONE. After careful consideration, we feel that it has merit but does not fully meet PLOS ONE’s publication criteria as it currently stands. Therefore, we invite you to submit a revised version of the manuscript that addresses the points raised during the review process. **In particular, your manuscript was evaluated by two expert reviewers, who found merit in your work, but also raised several issues. These issues included statistics and the interpretation of data obtained from alcohol-naive rats and were relevant to PLOS publication criteria n. 3 (experiments and statistics are performed to a high technical standard and described in sufficient detail) and n. 4 (conclusions are presented in an appropriate fashion and are supported by the data). I broadly agree with the Reviewers' comments, and encourage you to revise your manuscript accordingly.**

We look forward to receiving your revised manuscript.

Kind regards and happy new year,

Alessandro Silvani, M.D., Ph.D.

Academic Editor

PLOS ONE

**Journal Requirements:**

This study was supported by the Giovanni Armenise Harvard Foundation Career Development Award to LdV, and the National Institutes of Health grant AA017447 to MR and RC.

Reviewers' comments:

Reviewer's Responses to Questions

**Comments to the Author**

1. Is the manuscript technically sound, and do the data support the conclusions?

Reviewer #1: Partly

Reviewer #2: Partly

2. Has the statistical analysis been performed appropriately and rigorously? 

Reviewer #1: I Don't Know

Reviewer #2: No

3. Have the authors made all data underlying the findings in their manuscript fully available?

Reviewer #1: Yes

Reviewer #2: Yes

4. Is the manuscript presented in an intelligible fashion and written in standard English?

Reviewer #1: Yes

Reviewer #2: Yes

5. Review Comments to the Author

**Reviewer #1:**  Major points:

1. Discussion could benefit from interpretation of the findings in relation to propensity for alcohol taking (e.g., how does potential effect on cognition, or learning and memory, impact alcohol drinking, or propensity to start drinking).

2. Given the msP rats are alcohol-naïve, some of the comparisons of results to sleep changes reported in individuals with AUD are not entirely applicable.

3. More information is expected to describe data acquisition software. Also, please make it clear whether data were individually hand scored or autoscored in batches. Autoscoring typically lacks reliability when compared to hand score.

4. Age of rats at time of recordings spans adolescence to adult? (average 58+/- 8 to 12 days). The range of age here is potentially problematic. Can data be separated and analyzed by age (although there may not a large enough group size for this and thus, additional data points for either age may be required). Sleep maturation during adolescence is an important factor to consider as sleep patterns undergo discernible changes including circadian rhythm shifts that could confound the early/late sleep measures here amongst other measures.

5. Why were sleep spindles only assessed during “early and late sleep,” or first and last 3 hours of the light period? what was happening in the other 6 hours? Why are these period of interest more than peak rest time?

Minor points for clarification:

1. If band pass on EEG was 0.1- 40 hz, why was beta power limited to 25 Hz?

2. For post hoc analyses, why fisher LSD in some cases and Sidak’s MCT in others?

3. Figure 1- “W” sometimes indicates Wistar and other times abbreviates “wake/waking,” please correct. For Figure 1A, figure says WAKE was n.s. but results text indicates significance- which is it? Statistical results are not detailed in Results for Figure 1C.

4. In results, the time is referred to as a percentage, but the graphs are showing time in minutes, please make them match.

5. Figure 4B, does not appear to be “change in uV2.”

**Reviewer #2: ** The manuscript by de Vivo et al. characterized the sleep phenotype of a preclinical rat model of patients with AUD. Overall this investigation was conducted adequately providing a baseline dataset for this specific rat model which, in future experiments, will be useful to deeper understanding the neural circuits at the basis of substance use vulnerability.

However, I have some major methodological and conceptual concerns:

- I believe this work would have been more complete if the authors had included groups exposed to alcohol (possibly using a longitudinal protocol). Limiting the sleep profiles of this model to baseline conditions is of course of interest, however, it would have been nice to see what specific sleep changes are due to alcohol exposure rather than genetic predisposition. Please comment on this.

- Similarly, the authors claim that the altered sleep profile in msP rats might entail memory consolidation and behavioral impairments. However, since they did not provide any data on these aspects, I suggest reporting these speculations in a specific limitation paragraph together with the reply of my previous point.

- I noticed the presence of outliers in several figures, and I feel that, at least some of them, importantly impacted on the statistics and the conclusion of the present work. In particular, fig. 1F (msp group), 1G (W-NREM, wistar group), 4C (msp group), 5B (w group). Did the authors check for the weight of them on their conclusions? Did they apply any automatic outlier detection?

- How did the authors correct for multiple comparisons in fig. 1c?

- Fig. 1D, the statistics in the text seems referring to a main ANOVA group effect rather than an interaction effect before applying the corrected post-hoc analysis.

I also have some minor concerns/suggestions:

- For figures 4E-4I, 5D-H, how did the authors choose to limit the SWA and spindle analyses to the first and last 3 hours of NREM sleep? Could you provide any reference for this choice?

- I would suggest including the sleep fragmentation index (n° of wakefulness episodes / total sleep time) to make easier (for the readers) the conclusions on this characteristic

- I would also suggest including representative hypnograms to readily appreciate the different sleep structure between groups.

6. PLOS authors have the option to publish the peer review history of their article (what does this mean? ). If published, this will include your full peer review and any attached files.

**Do you want your identity to be public for this peer review?** For information about this choice, including consent withdrawal, please see our Privacy Policy .

Reviewer #1: No

Reviewer #2: No

---

## [Author Response · Author response to Decision Letter 1]

12 Feb 2025

Dear Dr. Silvani,

we have addressed the points raised by the reviewers, which are now included in a revised version of the manuscript. In light of these changes, we believe that the manuscript is much stronger in the current form and we ask you to consider it for publication.

In the submission files, we include a clean revised manuscript, supplementary figures S1-S3, supplementary Table S1, along with a point-to-point reply letter to the reviewers, with tables and figures regarding the outlier analysis and the statistical details asked by the reviewers. We kindly ask you to refer to the attached file for our response to the reviewers' specific comments.

Moreover, we declare that this study was supported by the Giovanni Armenise Harvard Foundation Career Development Award to LdV, and the National Institutes of Health grant AA017447 to MR and RC. There was no additional external funding received for this study.

Thank you.

Best regards,

Luisa de Vivo

---

## [Decision Letter · Decision Letter 1]

6 Apr 2025

PONE-D-24-48364R1Sleep phenotyping in a rat model of susceptibility to substance use disordersPLOS ONE

Dear Dr. de Vivo,

Thank you for submitting your manuscript to PLOS ONE. After careful consideration, we feel that it has merit but does not fully meet PLOS ONE’s publication criteria as it currently stands. Therefore, we invite you to submit a revised version of the manuscript that addresses the points raised during the review process. **Specifically, I agree with Reviewer 1 that statistical issues were not adequately addressed in the first revision of your manuscript. Moreover, the Grubbs test is designed to identify single outliers from relatively normal distributions, which may have not been always the case in your dataset, and ANOVA is sensitive to non-normal distributions as well as to outliers. On the other hand, your results seem quite robust to different statistical approaches, and I understand your choice to provide a full representation of your results in the main manuscript. On these bases, I recommend that you perform a further minor revision of the analysis and presentation of your results to fully address these issues, possibly seeking the advice of a professional biostatistician. You may consider consistently using non-parametric tests instead of parametric tests to decrease the sensitivity of results to potential outliers. Please provide a detailed description of your statistical approach, including the identification and management of outliers, in the methods section of the main manuscript, and discuss any related limitation in the discussion section.** Please submit your revised manuscript by May 21 2025 11:59PM. If you will need more time than this to complete your revisions, please reply to this message or contact the journal office at plosone@plos.org . Please include the following items when submitting your revised manuscript:

We look forward to receiving your revised manuscript.

Kind regards,

Alessandro Silvani, M.D., Ph.D.

Academic Editor

PLOS ONE

**Journal Requirements:**

Reviewers' comments:

Reviewer's Responses to Questions

**Comments to the Author**

1. If the authors have adequately addressed your comments raised in a previous round of review and you feel that this manuscript is now acceptable for publication, you may indicate that here to bypass the “Comments to the Author” section, enter your conflict of interest statement in the “Confidential to Editor” section, and submit your "Accept" recommendation.

Reviewer #1: (No Response)

Reviewer #2: All comments have been addressed

2. Is the manuscript technically sound, and do the data support the conclusions?

Reviewer #1: Yes

Reviewer #2: Yes

3. Has the statistical analysis been performed appropriately and rigorously? 

Reviewer #1: Yes

Reviewer #2: Yes

4. Have the authors made all data underlying the findings in their manuscript fully available?

Reviewer #1: Yes

Reviewer #2: Yes

5. Is the manuscript presented in an intelligible fashion and written in standard English?

Reviewer #1: Yes

Reviewer #2: Yes

6. Review Comments to the Author

**Reviewer #1: ** The authors have adequately addressed this reviewer's concerns, in large part, and the manuscript is much improved from the previous version. It is not clear, however, why the authors chose to present data both with (main figures) and without (supplemental figure S3) outliers. This reviewer would prefer the authors present only the figures and statistics that exclude outliers in the main body, and possibly only include figures with outliers as a supplement if they feel it necessary to show both. Reporting a statistically significant result that hinges on an outlier does little to enhance the reader's interpretation of the data.

**Reviewer #2:**  I have no furhter comments and I agree with the authors that the outlier analyses can be kept in the supplementary material.

7. PLOS authors have the option to publish the peer review history of their article (what does this mean? ). If published, this will include your full peer review and any attached files.

**Do you want your identity to be public for this peer review?** For information about this choice, including consent withdrawal, please see our Privacy Policy .

Reviewer #1: No

Reviewer #2: **Yes: ** Stefano Bastianini

---

## [Author Response · Author response to Decision Letter 2]

14 Apr 2025

Dear Editor,

we have addressed the points raised by the reviewers and the Academic Editor concerning the statistical analysis and representation of results. Please find our detail answer in the Response to Reviewers file and in the revised versions of the Manuscript, Figures, and Supplementary Table S1.

Best regards,

Luisa de Vivo

---

## [Editor Report · Decision Letter 2]

25 Apr 2025

Sleep phenotyping in a rat model of susceptibility to substance use disorders

PONE-D-24-48364R2

Dear Dr. de Vivo,

We’re pleased to inform you that your manuscript has been judged scientifically suitable for publication and will be formally accepted for publication once it meets all outstanding technical requirements.

Kind regards,

Alessandro Silvani, M.D., Ph.D.

Academic Editor

PLOS ONE
---

## [Editor Report · Acceptance letter]

PONE-D-24-48364R2

PLOS ONE

Dear Dr. de Vivo,

I'm pleased to inform you that your manuscript has been deemed suitable for publication in PLOS ONE. Congratulations! Your manuscript is now being handed over to our production team.

Kind regards,

on behalf of

Prof. Alessandro Silvani

Academic Editor

PLOS ONE